# Preparation and Research of a High-Performance ZnO/SnO_2_ Humidity Sensor

**DOI:** 10.3390/s22010293

**Published:** 2021-12-31

**Authors:** Fan Li, Peng Li, Hongyan Zhang

**Affiliations:** 1Xinjiang Key Laboratory of Solid State Physics and Devices, Xinjiang University, Urumqi 830046, China; 2School of Physical Science and Technology, Xinjiang University, Urumqi 830046, China; fan_li1995@163.com (F.L.); zhy@xju.edu.cn (H.Z.)

**Keywords:** zinc oxide/tin dioxide, specific surface area, oxygen vacancy, high response, humidity sensor

## Abstract

A high-performance zinc oxide/tin dioxide (ZnO/SnO_2_) humidity sensor was developed using a simple solvothermal method. The sensing mechanism of the ZnO/SnO_2_ humidity sensor was evaluated by analyzing its complex impedance spectra. The experimental results prove that the ZnO/SnO_2_ composite material has a larger specific surface area than pure SnO_2_, which allows the composite material surface to adsorb more water to enhance the response of the ZnO/SnO_2_ humidity sensor. ZnO can also contribute to the generation of oxygen-rich vacancies on the ZnO/SnO_2_ composite material surface, allowing it to adsorb a large amount of water and rapidly decompose water molecules into conductive ions to increase the response and recovery speed of the ZnO/SnO_2_ humidity sensor. These characteristics allowed the Z/S-2 humidity sensor to achieve a higher response (1,225,361%), better linearity, smaller hysteresis (6.6%), faster response and recovery speeds (35 and 8 s, respectively), and long-term stability at 11–95% relative humidity. The successful preparation of the ZnO/SnO_2_ composite material also provides a new direction for the design of SnO_2_-based resistance sensors with high humidity-sensing performance.

## 1. Introduction

Humidity sensors play a very important role in many fields, such as those related to the atmospheric environment, medical equipment, agricultural cultivation, aerospace, and industrial production [1,2,3]. The demand for a humidity sensor with higher response, faster response and recovery speed, better selectivity, and better stability and repeatability than currently available commercial sensors has recently increased [4,5,6]. To meet these demands, many types of humidity sensors have been designed, such as those that use surface acoustic waves [7], resistance [8], quartz crystal microbalance [9], field-effect transistors [10], optical fibers [11], or capacitance [12]. Among these humidity sensors, resistive humidity sensors are widely manufactured and applied because of their suitable linearity, high response, suitable working stability, relatively simple structure, and low cost [13]. Regarding resistive humidity sensors, the type of sensitive material remains to be the key to determining its performance. In recent years, polymers [14], metal oxide semiconductors [15,16], and carbon nanostructures [17,18] have often been used to manufacture resistive humidity sensors. Among these materials, metal oxide semiconductor materials are commonly used as humidity-sensitive materials because of their relatively large specific surface area, simple manufacturing process, and low cost [19,20]. As a typical metal oxide semiconductor material, SnO_2_ has recently become viewed as a potential humidity-sensing material owing to its wide band gap, relatively simple structure, suitable selectivity, and low cost [20,21]. In addition, SnO_2_ also exhibits suitable electrochemical stability, a reasonable specific surface area, and oxygen vacancies [1]. The specific surface area can adsorb water, and the oxygen vacancies on the surface of the sensitive materials can dissociate water molecules into conductive ions to increase the performance of humidity sensors. However, because the performance of pure SnO_2_ humidity sensors is affected by the specific surface area and oxygen vacancies, these sensors have exhibited low response, poor linearity, and long response and recovery times [1,22].

To further improve the performance of SnO_2_ humidity sensors, doping and compounding methods are often used. ZnO is an n-type oxide semiconductor material that is often applied for the development of humidity sensors owing to its high thermal and chemical stability, low cost, large specific surface area, and high electrical conductivity [13]. The formation of ZnO and SnO_2_ composite materials can promote the generation of more surface oxygen vacancies and a larger specific surface area. ZnO/SnO_2_ composite materials have a larger specific surface area, which allows more water to be adsorbed on the surface of the composite material to improve the response of the ZnO/SnO_2_ humidity sensor. Moreover, rich oxygen vacancies are generated on the ZnO/SnO_2_ composite material surface, which can adsorb many water molecules and decompose a large amount of water to form H_3_O^+^ to increase the conductivity of the composite material [23,24], thus shortening the response time of the ZnO/SnO_2_ humidity sensor.

In this study, a ZnO/SnO_2_ humidity sensor is prepared using a simple solvothermal method. The humidity-sensing mechanism is evaluated by analyzing the complex impedance spectrum (CIS). The results revealed that the ZnO/SnO_2_ composite material has a relatively large specific surface area that allowed it to absorb a large amount of water to improve the response of the humidity sensor. Many oxygen vacancies were observed on the ZnO/SnO_2_ composite material surface, which was found to be able to adsorb water and quickly dissociate many water molecules into H_3_O^+^; these characteristics served to improve the response and recovery speed of the sensor. For these reasons, the ZnO/SnO_2_ humidity sensor was able to achieve a higher response, faster response and recovery speed, better linearity, and lower hysteresis between 11% relative humidity (RH) and 95% RH.

## 2. Experimental

### 2.1. Materials

Zinc acetate ((CH_3_COO)_2_Zn·2H_2_O), tin (IV) chloride dehydrate (SnCl_4_·5H_2_O), urea (CH_4_N_2_O), ethanol (C_2_H_5_OH), and hexamine (C_6_H_12_N_4_) were purchased from Sinopharm Chemical Reagent Co., Ltd., (Tianjin, China). Ammonia (NH_3_·H_2_O) was purchased from Aladdin (Shanghai, China), and 3,5-dinitrosalicylic acid (C_7_H_4_N_2_O_7_) was purchased from Sinopharm Chemical Reagent Co., Ltd. (Shanghai, China). All of the chemical reagents used in our experiment were of analytical grade and could be used directly as received; additionally, deionized (DI) water was used in all of the experiments.

### 2.2. Synthesis of ZnO

ZnO was prepared via a solvothermal method, as shown in Figure 1. First, 0.554 g of (CH_3_COO)_2_Zn·2H_2_O was dissolved in 25 mL of DI water and magnetically stirred for 30 min to form Solution A. Second, 0.354 g of C_6_H_12_N_4_ was dissolved in 25 mL of DI water and magnetically stirred for 5 min to form Solution B. Solution B was then poured into Solution A, and the mixture was magnetically stirred for 30 min until evenly mixed. NH_3_·H_2_O was added to the mixture to obtain a pH of 10. Then, the mixture was poured into a 100 mL Teflon-lined autoclave, sealed, and placed in a drying oven at 140 °C for 4 h. After the reactant was naturally cooled to room temperature, it was washed with DI water and ethanol three times and then dried at 60 °C for 8 h.

### 2.3. Synthesis of ZnO/SnO_2_ Composites

ZnO/SnO_2_ composite materials were synthesized via a simple solvothermal method, as shown in Figure 1. First, 1.753 g of SnCl_4_·5H_2_O and 1.502 g of CH_4_N_2_O were added to 50 mL of DI water under magnetic stirring for 5 min to form a homogeneous solution. Then, 0.571 g of C_7_H_4_N_2_O_7_ was added to the solution and thoroughly stirred. After 10 min, 0 g, 0.081 g, 0.041 g, 0.027 g, and 0.02 g of ZnO were separately added to the above-described solution. The mixtures were continuously stirred in a water bath at 100 °C for 5 h. After the reactants cooled to room temperature, they were washed with DI water and ethanol three times before being dried at 60 °C for 8 h to obtain SnO_2_ and the ZnO/SnO_2_ composite materials. The composite materials are denoted as Z/S-1, Z/S-2, Z/S-3, and Z/S-4, respectively.

### 2.4. Characterizations

The phases of the samples were analyzed using D8 Advance X-ray diffraction (XRD) (Bruker, Karlsruhe, Germany). The morphology and images of the samples were identified using SU8010 scanning electron microscopy (SEM) (Hitachi, Japan) and JEM-2100 transmission electron microscopy (TEM) (Hitachi, Japan). Additionally, UV-vis spectra were obtained using a Lambda 650 UV-vis spectrophotometer (PerkinElmer, New York, NY, USA). The surface compositions and elemental states were analyzed using ESCALAB 250Xi X-ray photoelectron spectroscopy (XPS) (Thermo Fisher Scientific, Waltham, MA, USA). The structures of the samples were identified using a VERTEX 70 Fourier transform infrared spectroscopy (FTIR) system (Bruker, Karlsruhe, Germany). Lastly, the specific surface area and pore diameter were determined with Brunauer–Emmett–Teller (BET) instrument (Micrometrities, ASAP 2460).

### 2.5. Design of Humidity Sensor

First, 1 mL of DI water and a small amount of each sample was poured into a mortar and ground to form a paste, which was applied to coat the surface of the Ag-Pd cross electrode. Then, the Ag-Pd cross electrode was placed in a drying oven at 60 °C for 12 h to form a humidity-sensing element. Humidity-sensing performance tests were performed using a Zennium X workstation (Zahner, Kronach, Germany), the test AC voltage was set to 1 V, and the test temperature was 25 °C. The RH environments were achieved using LiCl, MgCl_2_, K_2_CO_3_, NaBr, NaCl, KCl, and KNO_3_ saturated salt solutions to obtain RH levels of 11%, 33%, 43%, 59%, 75%, 85%, and 95%, respectively. The humidity sensors were placed into the different RH environments for testing, and the testing process is illustrated in Figure 2. The response or recovery time of a humidity sensor is defined as the time required for the change in sensor impedance to reach 90% of the total impedance change.

## 3. Results and Discussion

### 3.1. Material Characterization

Figure 3 shows the XRD patterns of SnO_2_, ZnO, and the Z/S-2 composite material; these patterns were used to analyze the crystallinity and phase structure of each of the samples. The patterns show that there were diffraction peaks distributed between 20° and 80°. The SnO_2_ 2θ diffraction peaks at 26.61°, 33.89°, and 51.78°, respectively, correspond to the lattice planes (110), (101), and (211), which are consistent with the tetragonal rutile structure of SnO_2_ (JCPDS card no. 41-1445) [25]. The ZnO 2θ diffraction peaks at 31.77°, 34.42°, 36.25°, 47.54°, 56.60°, 62.86°, 66.38°, 67.96°, 69.10°, and 76.95° respectively correspond to the (100), (002), (101), (102), (110), (103), (200), (112), (201), and (202) lattice planes, which is indicative of the hexagonal wurtzite ZnO structure (JCPDS card no. 36-1451) [26]. The diffraction peaks for the Z/S-2 composite material were the same as those for SnO_2_, but the characteristic diffraction peak of ZnO was not detected; this may be because of the lower content of ZnO in the composite.

SEM images of the SnO_2_, ZnO, and Z/S-2 composite materials and TEM images of the Z/S-2 composite material were analyzed to further characterize their surface morphologies and microstructures, as shown in Figure 4. As can be seen in Figure 4a, SnO_2_ appeared as particles with a diameter of approximately 1 μm, and a large amount of agglomeration was observed. As shown in Figure 4b, ZnO appeared as rod-like structures with a length of approximately 5 μm. In the case of the Z/S-2 composite material, SnO_2_ appeared as an irregular block, and ZnO was uniformly attached to the surface of SnO_2_ (Figure 4c). The Z/S-2 composite material was observed to have a larger surface area and rougher surface than SnO_2_; these attributes served to create a number of adsorption sites and allow the adsorption of a large amount of water, which improved the sensor response [27]. Figure 4d,e show TEM images of the Z/S-2 composite material at low and high magnifications, respectively. From Figure 4d, it is clear that the Z/S-2 composite material had a block structure, indicating consistency with the SEM test results. Figure 4e shows the lattice fringes of SnO_2_ and ZnO; it can be seen that the lattice spacings of 0.14 and 0.34 nm correspond to the (200) crystal plane of ZnO, and the (110) crystal plane of SnO_2_, respectively. Based on these observations, ZnO can be considered to have been successfully compounded with SnO_2_.

Figure 5 shows the UV-vis absorption spectra and band gap energies of SnO_2_, ZnO, Z/S-1, Z/S-2, Z/S-3, and Z/S-4. As can be seen in Figure 5a, all sample types exhibited strong absorption in the ultraviolet region; particularly, the absorption edge appeared at 488.7 nm, 448.7 nm, 516.2 nm, 528.7 nm, 531.8 nm, and 512.2 nm, respectively. The absorption edges of the ZnO/SnO_2_ composite materials were observed at higher wavelengths than that of SnO_2_, indicating that the combination of ZnO and SnO_2_ reduced the band gap energy of SnO_2_. Figure 5b shows the band gap energies of all of the sample materials; note that the band gap energy was calculated using the Kubelka–Munk equation ${\left(\alpha h\nu \right)}^{2}=A\left(h\nu -{Ε}_{g}\right)$, where α is the absorption coefficient, hν is the photon energy, A is a constant, and ${Ε}_{g}$ is the band gap energy [25]. According to the results shown in Figure 5b, the band gap energies of SnO_2_, ZnO, Z/S-1, Z/S-2, Z/S-3, and Z/S-4 were 2.97 eV, 3.15 eV, 2.63 eV, 2.61 eV, 2.58 eV, and 2.73 eV, respectively. The results indicate that ZnO recombination with SnO_2_ can gradually reduce the band gap energy of SnO_2_ and that the band gap energy increases as the ZnO content increases. As is well known, a smaller band gap in the ZnO/SnO_2_ composite materials is more favorable for electron transfer in semiconductors; this is a reason why the composite material improves the response of the humidity sensor.

The elemental chemical states and compositions of the SnO_2_ and Z/S-2 composite materials were measured using XPS; the results are shown in Figure 6. The Sn 3d peaks for the SnO_2_ and Z/S-2 composite material are shown in Figure 6a. The binding energies of the Sn 3d_5/2_ peak were 486.7 and 486.9 eV, respectively, and those of the Sn 3d_3/2_ peak were 495.1 and 495.3 eV, respectively. The difference between the binding energy of Sn 3d_3/2_ and Sn 3d_5/2_ was 8.4 eV, indicating the presence of Sn^4+^ in the samples [28,29,30]. The position of the Sn 3d peak for the Z/S-2 composite material shifted relative to the position of the peak associated with SnO_2_. In particular, the peak shifted toward higher binding energy, which implies that the presence of Zn^2+^ increased the binding energy of the Z/S-2 composite material [31]. As shown in Figure 6b, the binding energies of the Z/S-2 composite material at the Zn 2p_3/2_ and Zn 2p_1/2_ peaks were 1021.2 and 1044.3 eV, respectively. The difference between the binding energy for the Zn 2p_1/2_ and Zn 2p_3/2_ peaks was 23.1 eV, which corresponds to the binding energy of Zn^2+^ in ZnO; this indicates that ZnO was present in the Z/S-2 composite material [28,32]. Figure 6c,d show the O 1s spectra of the SnO_2_ and Z/S-2 composite materials. The O 1s peak can be separated into three components: lattice oxygen (O_1_), oxygen vacancies (O_2_), and chemisorption dissociation oxygen (O_3_) [31]. Table 1 shows the locations of the respective peaks and percentages of the three oxygen components in the O 1s spectra of the SnO_2_ and Z/S-2 composite materials. Three SnO_2_ O 1s peaks had binding energies of 530.65, 531.80, and 532.65 eV, while the three Z/S-2 composite O 1s peaks had binding energies of 530.71, 531.95, and 532.60 eV. The percentages of O_2_ in the SnO_2_ and Z/S-2 composite material were 26.83% and 35.22%, respectively. The O_2_ area of the Z/S-2 composite material was significantly larger than that of SnO_2_. It is well known that oxygen vacancies promote water adsorption on the surface and the decomposition of water molecules [13]. A large number of oxygen vacancies in Z/S-2 can lead to the adsorption of a large amount of water and accelerate the decomposition of water molecules into conductive ions, thus reducing the sensor response time and enhancing the humidity-sensing performance.

The chemical bonds of the SnO_2_, ZnO, and Z/S-2 composite material were analyzed using FTIR; the results are shown in Figure 7. In the case of ZnO, there were three peaks at 521.18, 1631.46, and 3411.776 cm^−1^, corresponding to the tensile vibration mode of Zn-O, the bending mode of water, and the stretching vibration mode of O-H, respectively [33]. Among them, O-H is a hydrophilic functional group that can adsorb water to enhance the response of the sensor. The FTIR spectra of the SnO_2_ and Z/S-2 composite were similar, and many absorption peaks were observed. At 544.76 cm^−1^, the tensile mode of the Z/S-2 composite material is Zn-O and Sn-O [34]. The absorption peak of the Z/S-2 composite material was wider than that of SnO_2_; furthermore, it was shifted to a lower wavenumber. Thus, it was confirmed that ZnO was successfully compounded with SnO_2_. Additionally, C-H and O-H stretching vibrations were present between 808.38 and 1279 cm^−1^ [35]. Moreover, Sn-OH and O-H bending vibration modes were observed at approximately 1300 and 1606.37 cm^−1^, respectively [25,34]. An N-H tensile vibration mode was also observed at 3092.54 cm^−1^, and an O-H tensile vibration mode was observed at 3450.99 cm^−1^ [36]. These modes represent hydrophilic functional groups that can absorb more water to enhance the humidity-sensing performance.

Figure 8 shows the N_2_ adsorption/desorption isotherms and BET pore diameter distribution of the ZnO, SnO_2,_ and Z/S-2 composite material. The BET-specific surface areas of these materials are 38.415, 24.674, and 56.485 m^2^/g, respectively. The average pore sizes of the ZnO, SnO_2,_ and Z/S-2 composite material are 6.59874, 8.89122, and 10.1003 nm, respectively. It is believed that a large specific surface area can improve the adsorption capacity of water molecules. Therefore, the Z/S-2 composite material exhibits a suitable humidity-sensing performance.

### 3.2. Sensor Characterization

Figure 9a shows the impedance data of the SnO_2_, ZnO, Z/S-1, Z/S-2, Z/S-3, and Z/S-4 humidity sensors as a function of RH under the condition of an operating frequency of 100 Hz. The linearity of the ZnO/SnO_2_ humidity sensor was found to be better than that of the SnO_2_ and ZnO humidity sensors, with the best linearity being observed for the Z/S-2 humidity sensor. The changes in the impedance of the ZnO and SnO_2_ humidity sensors were found to be approximately two orders of magnitude, whereas that of the Z/S-2 humidity sensor was more than four orders of magnitude. The response of a humidity sensor can be defined as $R=\frac{R_{L}-R_{H}}{R_{H}}\times 100\%$ [1], where $R_{L}$ is the impedance value at 11% RH, and $R_{H}$ is the impedance value at 95% RH. The responses of the SnO_2_, ZnO, Z/S-1, Z/S-2, Z/S-3, and Z/S-4 humidity sensors were calculated to be 40,906%, 6441%, 434,113%, 1,225,361%, 59,983%, and 20,566%, respectively. The Z/S-2 humidity sensor had the largest response, which occurred primarily because of the compounding of ZnO, resulting in a change in the number of oxygen vacancies on the SnO_2_ surface. The Z/S-2 humidity sensor had the most oxygen vacancies, and abundant oxygen vacancies have been reported to dissociate a large number of water molecules into H_3_O^+^, resulting in increased conductivity and an improved sensor response [13]. Figure 9b shows the response and recovery characteristics of the Z/S-2 humidity sensor at different RH levels (the RH changes were completed within 1 s). It can be observed that the impedance change of the Z/S-2 humidity sensor at different RHs remained relatively constant, indicating that the sensor has suitable repeatability. Figure 9c shows that the impedance of the Z/S-2 humidity sensor changes as the RH is varied. As RH is increased, the impedance decreases, indicating that the Z/S-2 humidity sensor has an excellent sensing performance. In Figure 9d, it can be observed that the drying temperature has an effect on the Z/S-2 humidity sensor. It is apparent that when the drying temperature was 60 °C, the response and linearity of the Z/S-2 humidity sensor was superior to that obtained at other drying temperatures.

Figure 10a shows the relationship between impedance and RH for the Z/S-2 humidity sensor while operating at different frequencies (40, 100, 1k, 10k, and 100k Hz). When the operating frequency was 1k, 10k, and 100k Hz, the change in impedance was not significant at low RH levels. This is because the water molecules absorbed by the Z/S-2 humidity sensor could not be easily polarized in a high-frequency electric field [1,12]. The Z/S-2 humidity sensor exhibited a suitable response at operating frequencies of 40 Hz and 100 Hz; however, at 40 Hz, the linearity was found to be poor, and the impedance change was unstable. Thus, the best operating frequency for the Z/S-2 humidity sensor is 100 Hz. Figure 10b shows the adsorption and desorption results for the Z/S-2 humidity sensor under the conditions of different RH levels. When the RH level is the same, the maximum difference in impedance during adsorption and desorption can be defined as hysteresis [37]. Hysteresis error can be determined by solving $\gamma H=\frac{∆H_{max}}{2F_{FS}}$, where $∆H_{max}$ is the maximum impedance difference between adsorption and desorption processes, and $F_{FS}$ is the full-scale output value [23]. In the case of the Z/S-2 humidity sensor, the maximum hysteresis was determined to occur at 43% RH, and the hysteresis error was calculated to be 6.6%. The characteristic curves of the adsorption process and desorption process were observed to be very similar; thus, the Z/S-2 humidity sensor can be considered to have suitable reversibility. Figure 10c shows the response function of the Z/S-2 humidity sensor when it is operated at 100 Hz. This function can be described as $Y=3.39\times 10^{7}e^{-x/30.57}-2.51\times 10^{5}$, where X is the RH and Y is the impedance value. The regression coefficient was calculated to be $R^{2}=0.945$, which confirms that the Z/S-2 humidity sensor exhibited suitable linearity.

Figure 11a shows the continuous response and recovery curve results for the Z/S-2 humidity sensor under the conditions of RH levels of 11% to 95%. The response and recovery curves were very similar over four cycles of testing, confirming that the Z/S-2 humidity sensor has suitable repeatability. Note that the response or recovery time of a humidity sensor is defined as the time required for the change in sensor impedance to reach 90% of the total impedance change [21,37]. To ensure test result accuracy, the conversion time between different RH levels was set to be within 1 s. The response time and recovery time of the Z/S-2 humidity sensor were calculated to be 35 s and 8 s, respectively [13,21,38]. The Z/S-2 humidity sensor showed a fast response and recovery speed, mainly due to the large specific surface area of the Z/S-2 composite material, which permits the fast adsorption of a large number of water molecules onto its surface. The large number of oxygen vacancy defects on the surface of the Z/S-2 composite material permits the quick decomposition of water molecules into H_3_O^+^, which may then participate in conduction, thereby improving the response speed of this humidity sensor. In Table 2, the performance of the Z/S-2 humidity sensor is compared with similar humidity sensors that have been previously reported. The results show that the Z/S-2 humidity sensor has a better sensing performance. As shown in Figure 11b, the impedance of the Z/S-2 humidity sensor was measured under the conditions of different RH levels over a period of 30 d to verify its stability. Consequently, the sensor impedance was found to be largely unchanged under the condition of a constant RH level. Thus, the Z/S-2 humidity sensor has suitable stability and can be used to measure the humidity of an environment over a long period of time. To ensure the accuracy of the Z/S-2 humidity sensor for RH measurements, four gases, namely NH_3_, C_3_H_6_O, H_2_O_2,_ and C_2_H_6_O, were selected for cross-sensitivity testing. As shown in Figure 11c, the responses (R) of the sensor toward these four gases were 3.031, 1.285, 0.748, and 0.501, respectively. The conditions used in these experiments were 100 ppm, and R was defined as R = (R_gas_ − R_air_)/R_air_, where R_gas_ is the impedance of the sensor contacting the gas to be measured and R_air_ is the impedance of the sensor being placed in the air. Therefore, the response of the Z/S-2 humidity sensor to the gas can be ignored.

Figure 12 shows the CIS and equivalent circuit (EC) of the Z/S-2 humidity sensor under the conditions of different RH levels; these results can explain the sensing mechanism of the Z/S-2 humidity sensor. The CIS of the Z/S-2 humidity sensor was measured over the operating frequency range of 40 to 100k Hz. At 11% RH, the CIS was a long straight line, and the corresponding EC was determined to be composed of a constant phase element (CPE), indicating that only a small amount of water was adsorbed to form a chemisorbed water layer [42]. It is generally difficult for ions in the chemisorbed water layer to transfer. This phenomenon results in the generation of an abnormally high impedance value, which causes the intrinsic impedance graph to become a straight line [43,44]. As can be seen in Figure 12a, the CIS curve obtained under the condition of 33% RH was bent downward, gradually forming a semicircle. The corresponding EC is known to consist of a parallel capacitor (C_f_) and resistor (R_f_) [43,44]. Under these conditions, a small amount of water molecules is adsorbed onto the surface via both physical and chemical adsorption. Physically adsorbed water molecules are bonded with hydrogen bonds, which makes it difficult for water molecules to move freely, and the movement of electric charges is restricted. In this process, the strong electrostatic field around the oxygen vacancies results in the water molecules becoming adsorbed onto the sensitive material to be ionized, and H^+^ jumps between adjacent water molecules to conduct electricity [42]. As can be seen in Figure 12a, the CIS curves obtained under the conditions of 43% and 59% RH had the appearance of a short line that appears at the end of the semicircle in the low-frequency region. The short line represents the Warburg impedance (Z_W_), and the corresponding EC is the original circuit is in series with Z_W_ [45]. As the RH was increased, a physical adsorption layer began to form on the chemical adsorption layer, and water molecules were increasingly decomposed into H_3_O^+^ and OH^-^. H_3_O^+^ then becomes the main carrier, and it can move freely in the physical adsorption layer. A Grotthuss (H_2_O + H_3_O^+^ = H_3_O^+^ + H_2_O) chain reaction is formed during this process [42]. Increasing the RH from 75% to 85%, followed by a subsequent increase to 95%, was found to result in the gradual disappearance of the observed semicircle in the high-frequency area. Additionally, the curve tail was found to become longer and more prominent in the low-frequency area. In the corresponding EC, the resistor (R_f_) is in series with Z_W_, where Z_W_ was determined to occupy a dominant position [43,44]. At this point, the surface of the Z/S-2 humidity sensor adsorbs a large amount of moisture, and the Grotthuss chain reaction is accelerated, thereby causing the impedance to rapidly decrease. This consequently enhances the humidity-sensing performance of the sensor.

## 4. Conclusions

A high-performance ZnO/SnO_2_ humidity sensor was designed, and the sensing mechanism of the ZnO/SnO_2_ humidity sensor was investigated by analyzing the CIS. The results show that the Z/S-2 composite material has a larger specific surface area and can provide more absorption sites; thus, more water can be adsorbed to speed up the response of the humidity sensor. Many oxygen vacancies were observed on the Z/S-2 surface; this facilitates the absorption of water to improve the response of the sensor; these oxygen vacancies also help to dissociate water molecules into H_3_O^+^ to increase the conductivity and improve the response and recovery speed of the sensor. Thus, the Z/S-2 humidity sensor has suitable linearity and stability, small hysteresis (6.6%), high response (1,225,361%), and short response and recovery times (35 s and 8 s) under the conditions of RH levels of 11% to 95%. The Z/S-2 humidity sensor developed in this study provides a new idea for the development of high-performance SnO_2_ humidity sensors.

## Figures and Tables

**Figure 1 sensors-22-00293-f001:**
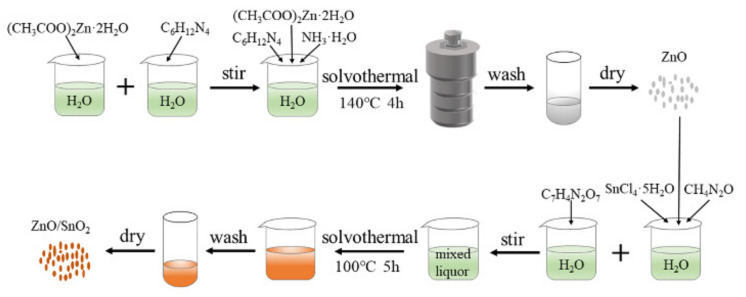
Preparation process of ZnO, SnO_2,_ and ZnO/SnO_2_ composites.

**Figure 2 sensors-22-00293-f002:**
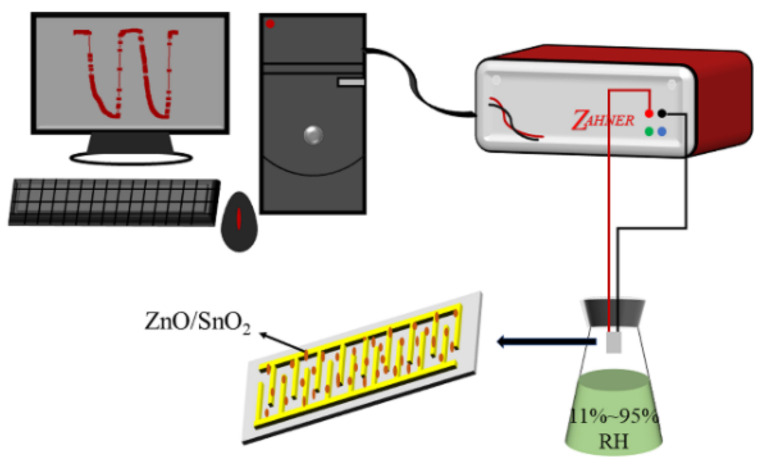
Testing process for the performance of ZnO/SnO_2_ humidity sensor.

**Figure 3 sensors-22-00293-f003:**
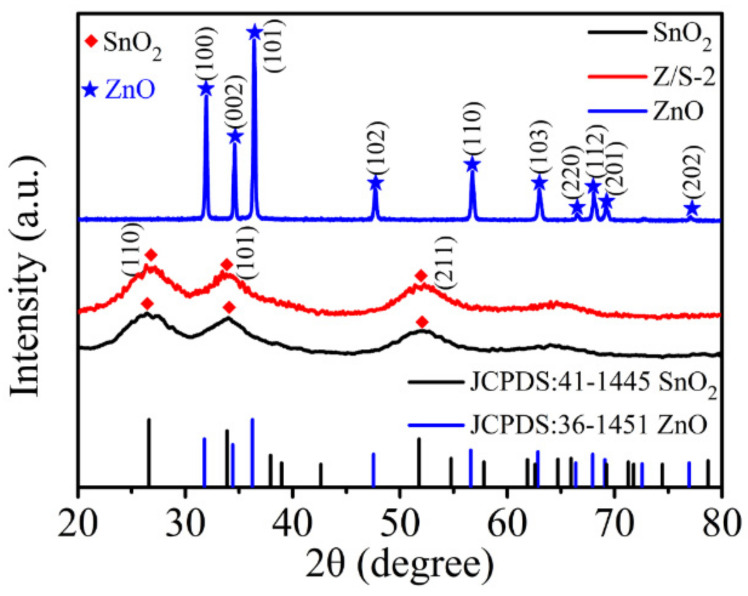
XRD patterns of ZnO, SnO_2_, and Z/S-2 composite.

**Figure 4 sensors-22-00293-f004:**
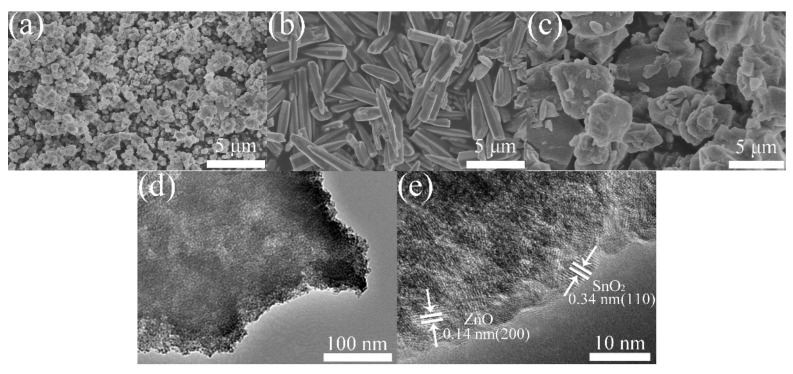
SEM spectra of (**a**) SnO_2_, (**b**) ZnO, and (**c**) Z/S-2 composite, and (**d**,**e**) TEM spectra of Z/S-2 composite.

**Figure 5 sensors-22-00293-f005:**
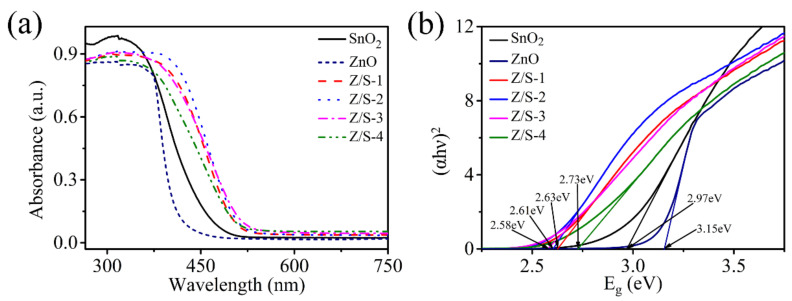
(**a**) UV-vis absorption spectra and (**b**) band gap energies of SnO_2_, ZnO, Z/S-1, Z/S-2, Z/S-3, and Z/S-4 composites.

**Figure 6 sensors-22-00293-f006:**
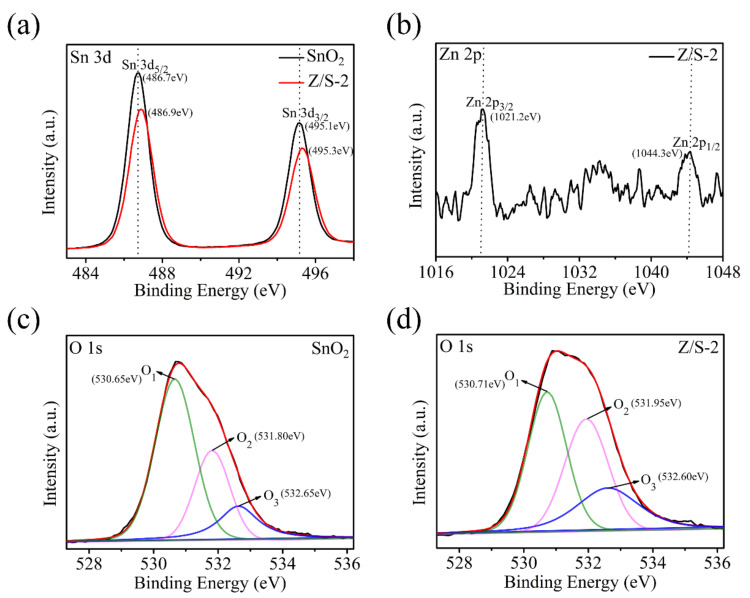
XPS spectra of (**a**) Sn 3d spectra of SnO_2_ and Z/S-2 composite, (**b**) Zn 2p spectrum of Z/S-2 composite, (**c**,**d**) O 1s spectra of SnO_2_ and Z/S-2 composite.

**Figure 7 sensors-22-00293-f007:**
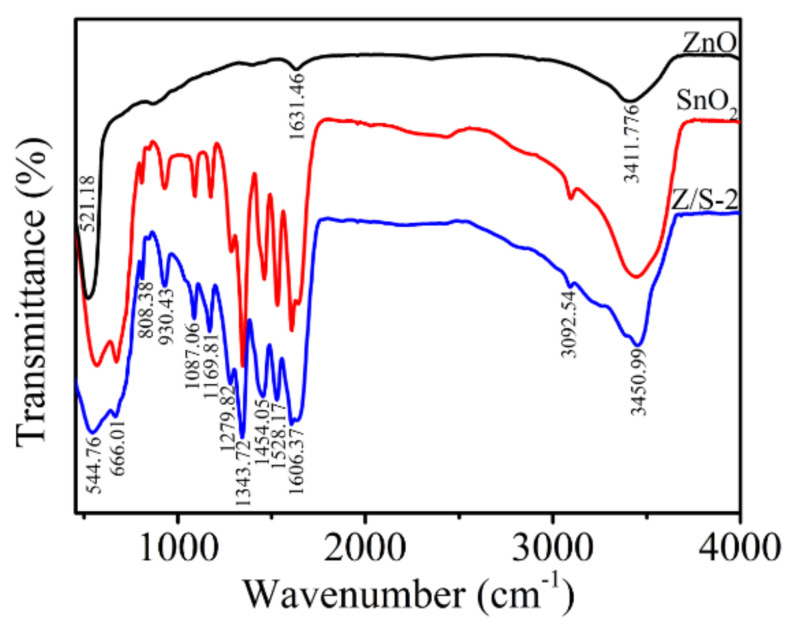
FTIR spectra of SnO_2_, ZnO, and Z/S-2 composite.

**Figure 8 sensors-22-00293-f008:**
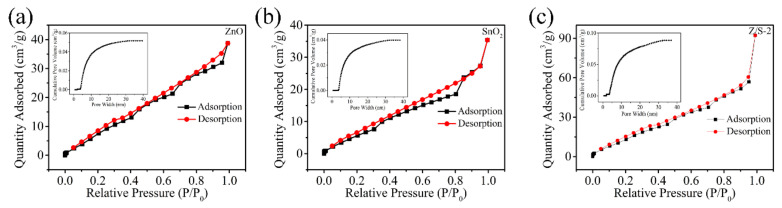
N_2_ adsorption-desorption isotherms and pore diameter distribution of the (**a**) ZnO, (**b**) SnO_2,_ and (**c**) Z/S-2 composite.

**Figure 9 sensors-22-00293-f009:**
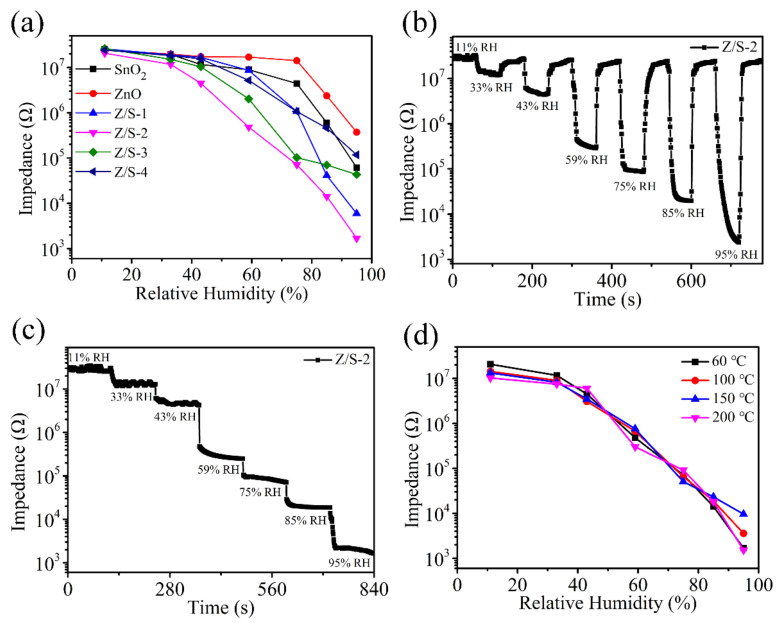
(**a**) Impedance changes of humidity sensors at 100 Hz, (**b**) response and recovery characteristics of Z/S-2 humidity sensor at different RHs, (**c**) impedance curve of Z/S-2 humidity sensor at different RHs, and (**d**) impedance response curves of Z/S-2 humidity sensor at drying temperatures of 60, 100, 150, and 200 °C.

**Figure 10 sensors-22-00293-f010:**
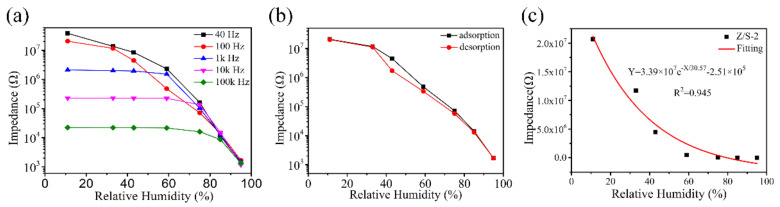
(**a**) Impedance changes of Z/S-2 humidity sensor at different operating frequencies, (**b**) hysteresis loop, and (**c**) response function of the Z/S-2 humidity sensor.

**Figure 11 sensors-22-00293-f011:**
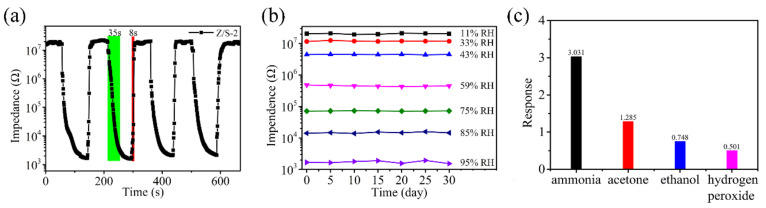
(**a**) Response/recovery characteristic curve of the Z/S-2 humidity sensor, (**b**) long-term stability under different relative humidity levels of the Z/S-2 humidity sensor and (**c**) responses of the Z/S-2 humidity sensor to ammonia (NH_3_), acetone (C_3_H_6_O), hydrogen peroxide (H_2_O_2_) and ethanol (C₂H_6_O).

**Figure 12 sensors-22-00293-f012:**
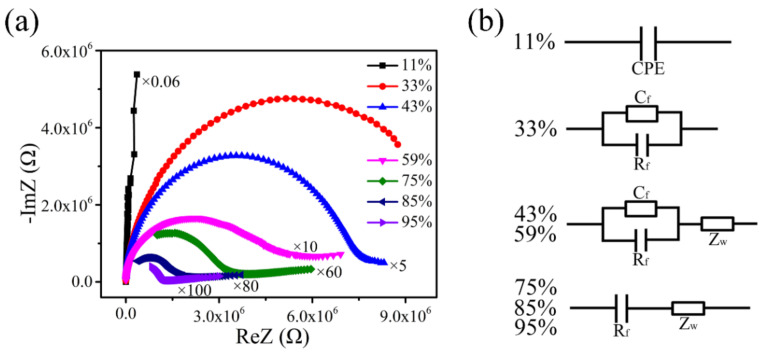
(**a**) Complex impedance spectra and (**b**) equivalent circuit of the Z/S-2 humidity sensor.

**Table 1 sensors-22-00293-t001:** The locations of the respective peaks and percentages of the three oxygen components in the O 1s spectra.

Sample	Oxygen Species	Binding Energy (eV)	Relative Percentage (%)
SnO_2_	O_1_	530.65	56.27
	O_2_	531.80	26.83
	O_3_	532.65	16.90
Z/S-2	O_1_	530.71	40.57
	O_2_	531.95	35.22
	O_3_	532.60	24.21

**Table 2 sensors-22-00293-t002:** A comparison of the humidity-sensing performance of the previously published work.

Material	Order of Impedance Change	Response Time (s)	Recovery Time (s)	Hysteresis (%)	Ref.
SnO_2_	2	32	25	--	[38]
SnO_2_/SiO_2_	2	14	16	2%	[39]
Fe/SnO_2_	0.8	--	--	--	[40]
NiO/SnO_2_	1	18.4	37.2	--	[41]
MoS_2_/SnO_2_	4	5	13	--	[1]
ZnO/SnO_2_	4	35	8	6.6%	This work

## Data Availability

Data sharing not applicable.

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
