# Peer review of "Preparation and Research of a High-Performance ZnO/SnO2 Humidity Sensor"

_sensors, 2021, doi:10.3390/s22010293_

Round 1
Reviewer 1 Report
Interesting work, but the paper needs more structure and a detailed discussion of the results in a conclusive way.
Introduction needs to address the novelty of the paper. What is novel with respect to the state of the art?
The structure of chapter 2 needs improvement. Materials, synthesis, characterization and humidity sensing are not clearly separated. Especially 2.5. Humidity sensing tests are unclear. Figure 1 should be separated into two figures and referred to in the appropriate section. The measurement setup for the impedance measurement is missing at all.
Chapter 3 Results: Should be split in 2 parts: Part 1 Material characterization and Part 2 Sensor characterization. The interpretation of the impedance results is not clear. What is the operating frequency? How was the impedance measured? How did you determine the response and recovery time? Explanations for the different equivalent circuits are missing. What does the impedance spectrum tell us?
Conclusions: OK
Reviewer 2 Report
Present manuscript by Li et al. deals with humidity sensing properties of ZnO/SnO2 composites. The authors used several techniques to characterize as-synthesized composites and facts regarding it are explained properly. In addition, the sensor based on ZnO/SnO2 composites shows an extraordinary response of 1225361%, which is considered huge in the sensing field.
Some minor issues:
1) Fig.3 labels are not visible. 2) regarding Fig. 7: Instead of a point graph, obtained impedance spectrums/curves are recommended for more clarity. 3) Response time of the sensors is higher when compared to the recovery time. Author's should explain this issue properly with suitable references.4) What is the motive behind humidity sensing? Did authors study sensing performances towards other oxidizing and reducing gases? 5) A comparison of sensing performance is recommended.
Reviewer 3 Report
The manuscript is well written and presented, however the author stands over the surface area effect over sensor response. Hence, BET measurements of the bare ZnO, SnO2 and composites are need to be compared and its effect over humidity response. Second the sample prepared were dried at 60oC which is very low and thus effect of temperature over sensing should be studied as function of humidity. Third, the role of oxygen vacancies and zinc interstitials need to be proved over humidity response which plays a major role in the sensor response.
Round 2
Reviewer 1 Report
Fig. 1 and Fig. 2 legends are mixed up.
Please correct accordingly.
There are many grammar mistakes (wrong use of tenses, word order,...) throughout the manuscript.
Please correct accordingly.
Reviewer 3 Report
The revised manuscript is very well written with an emphasis on the role of oxygen vacancies of composites towards humidity sensing. BET measurement data provides further experimental evidence of relating surface area with oxygen vacancies towards humidity sensing. Now the manuscript is very well integrated and be suitable for publishing.
